# Drivers and effects of fish-for-sex related single parenthood in a fishing coastal community in Ghana

Sylvester Kyei-Gyamfi[1⊙*], Frank Kyei-Arthur[2⊙*]

1 Department of Children, Ministry of Gender, Children and Social Protection, Accra, Ghana,
2 Department of Environment and Public Health, University of Environment and Sustainable Development, Somanya, Ghana

⊙ These authors contributed equally to this work.
* fkyei-arthur@uesd.edu.gh (FK-A); sylvesterr_kyeigyamfi@yahoo.com (SK-G)

## Abstract

The migration of fishers from one community to another is often associated with fish-for-sex (FFS) exchanges. FFS can lead to social issues such as absentee parenting and the generational cycle of FFS relationships. This study examined the perception of the frequency of FFS-related female single parenthood and the drivers and effects of single parenthood arising from FFS relationships in Elmina, Ghana. This study used a convergent parallel mixed-methods design to examine the views of 385 fishers, 30 key informants, and 20 focus group participants on FFS-related single parenthood. Descriptive statistics was used to analyze the quantitative data, while the transcript of participants were analyzed thematically. The findings show that most fishers (63.1%) indicated the occurrence of FFS female single parenthood in Elmina. The driver of FFS female single parenthood included uncertain paternity resulting from multiple sexual partners and male partners denial of paternity due to suspicions of promiscuity. Also, the adverse effects of FFS female single parenthood included paternal absence, child developmental challenges, maternal burden, and the intergenerational cycle of FFS relationships. This study demonstrated that FFS female single parenthood is a common phenomenon in Elmina. There is an urgent need for policymakers to design interventions to address the phenomenon of FFS female single parenthood to enhance the well-being of children and mothers with children from FFS relationships.

## Introduction

The fisheries sector plays a crucial role in the livelihoods of many communities, particularly in small-scale fisheries (SSF), which are characterized by their reliance on local resources and traditional fishing practices [1,2]. Globally, small-scale fishers operate in some of the biologically richest and most productive waters on earth, often

**Data availability statement:** All relevant data are within the paper and its Supporting Information files.

**Funding:** The author(s) received no specific funding for this work.

**Competing interests:** The authors have declared that no competing interests exist.

in tropical coastal zones. These fisheries are vital for food security, economic stability, and cultural identity within coastal communities in Ghana and across West Africa [1–3]. In Ghana, small-scale fishing has a long history dating back to the 1700s and 1800s, characterized by the use of purse seine nets, beach seine nets, set nets, drifting gill nets, and hooks and lines, all operated from dug-out canoes [4,5]. However, they often face significant challenges, including a lack of political support and recognition, which disproportionately affects women in the sector [1,6].

Most marine fishers do not limit their work to a single fishing community due to seasonal fish movements, tidal patterns, and economic opportunities [7–9]. Male fishers in Ghana's small-scale fisheries sector typically move between fishing communities to perform offshore fishing activities (such as catching fish at sea or working on boats and other equipment). In contrast, women often perform near-shore fishing activities, engaging in fish processing, financing fishing expeditions, and managing trade networks, which are crucial for sustaining the fisheries economy [6,10–14]. This interdependence underscores the critical roles each gender plays in sustaining the community's livelihood, fostering collaboration and mutual support as they depend on each other's skills and resources to succeed. For instance, in Ghana's industrial tuna value chain, women are not only traders but also vessel owners, financiers, and key decision-makers, illustrating the complexity of gendered roles in fisheries [6,15–17].

However, the mobility of fishers in their work is often associated with risky sexual relationships known as "fish-for-sex" (FFS) exchanges, which are prevalent in certain fishing communities in developing countries particularly in sub-Saharan Africa [18,19]. FFS, a transactional sexual practice, is prevalent in various fishing communities across Africa and Asia, where men and women trade fish for sexual favors [20,21]. For instance, studies have documented FFS in Nigeria [22], Kenya [19], Zambia [23], Malawi [21], and Ghana [24]. Fish-for-sex (FFS) arises from women's need for access to fish within gendered power structures. Male fishers exploit this need, often under unequal terms, leading women to engage in unprotected sex out of fear of losing access to fish [24–26]. Understanding this context is essential for addressing the broader implications of FFS on women's health and economic security.

FFS often leads to an imbalance, with either many men sharing a single woman or many women sharing one man [24]. Risky sexual behavior among fishers, such as multiple sexual partners and non-condom use [26–28], can lead to pregnancy and increase the risk of sexually transmitted infections for both fishers and their partners. These partners often include female fish traders, processors, and other women engaged in fisheries-related activities, whose economic dependence on male fishers may limit their ability to negotiate safer sexual practices. Also, both female and male fishers who engage in sexual intercourse with multiple partners may be uncertain about the paternity of their children [29]. Furthermore, FFS can lead to broader social issues such as absentee parenting and the generational impact on young people who often engage in FFS, having been introduced to it from an early age in communities where it is normalized.

Despite its far-reaching adverse impact on parenting, this issue has received limited attention in fisheries literature, particularly in Ghana. There is little comprehensive documentation on how FFS leads to women having children whose fathers are untraceable – either due to men's reluctance to take responsibility, citing promiscuity or because the male partners have moved to other fishing locations. This study aimed to examine the perception of the frequency of FFS-related female single parenthood and the drivers and effects of single parenthood arising from FFS relationships within the Elmina fishing community in the Komenda-Edina-Eguafo Abrem (KEEA) Municipality, Central region of Ghana. By doing so, it seeks to provide a deeper understanding of these complex dynamics.

## Materials and methods

### Study design and sampling procedure

The study employed a convergent parallel mixed-methods design to analyze the quantitative and qualitative data comprehensively. This design allows researchers to gather both quantitative and qualitative data simultaneously to compare, interrelate or validate the findings [30]. By integrating quantitative and qualitative data, researchers gain deeper insights and a richer understanding of complex issues. In a convergent parallel mixed-methods design, as outlined by Creswell and Clark [31], data collection and analysis for both quantitative and qualitative components occur concurrently in a single phase. Qualitative methods allowed for an in-depth examination of both male and female fishers mobility, safety concerns, and experiences of sexual harassment, while quantitative methods facilitated the analysis of relationships between the study's key variables.

**Quantitative sampling procedure.** The number of fishers in Elmina is unknown due to the lack of national statistics on small-scale artisanal fishers. Therefore, the sample size for the quantitative data was calculated using the formula below:

$$n = \frac{(Z-score)^2 * SD(1-SD)}{(Margin\ of\ error)^2} \tag{1}$$

n = sample size, Z score for 95% confidence interval = 1.96, SD is standard deviation = 0.5, and margin of error = 0.05.

$$n = \frac{(1.96)^2 * .5(1-0.5)}{(0.05)^2} \tag{2}$$

$$n = 384.16$$

From the calculation, the sample size was 384. However, the first author conveniently estimated the sample size to be 385. Consequently, a total of 385 fishers were interviewed. With the assistance of the chief fisherman and several contacts within the Elmina fishing community, the first author identified 10 fishers' associations. These associations represent a variety of groups, including boat owners, male fish traders, female market queens who owned boats and made substantial investments in the fishing business, as well as porters of fish and boat repairers. This diversity underscores the multifaceted nature of the fishing industry in Elmina. After compiling the list of all members from these ten associations, a total of 690 members was identified. A simple random sampling procedure was employed to identify 385 study respondents.

**Qualitative sampling procedure.** For the qualitative data, a purposive sampling procedure was used to select fishers and key informants. Thirty (30) key informant interviews (KIIs) were interviewed (See Table 1). Also, two (2) focus group discussions (FGDs) were conducted with ten (10) participants for each group. In total, twenty fishers participated in the FGDs. The FGDs consisted of two groups: male fishers and female fish traders. Participants in groups were chosen from

**Table 1. Participants of the key informant interviews.**

| Subject | Male | Female | Frequency |
|---|---|---|---|
| KEEA Administration Office (Coordinating Director, Assistant Planning officer, Municipal Director of Social Welfare and Community Development & Gender Desk Officer of the Assembly) | 3 | 1 | 4 |
| Municipal Health Directorate | 0 | 1 | 1 |
| Ghana AIDS Commission | 1 | 0 | 1 |
| Non-governmental or Community-based Organisations officials | 0 | 2 | 2 |
| Department of Fisheries | 0 | 2 | 2 |
| Community members (fisher associations, and other opinion leaders of the fishing community | 12 | 8 | 20 |
| Total | 16 | 14 | 30 |

the ten fisher associations, five of which represented male fishers and the other five representing female fish traders. Two fishers from each association were chosen to participate. Participants for the FGD were chosen based on age, residency status, and the ability to offer detailed information about fishing-related activities in Elmina. This approach ensured a diverse group that sufficiently represented a range of experiences essential for the study's objectives.

The study population comprised artisanal marine water fishers aged 18 or older who were actively involved in fishing, fish transport, boat repair, fishing gear sales, and fish trade. The sample size was calculated to ensure accurate proportion estimates while considering financial constraints, survey distribution time, and the undefined number of fishers in the area. Participants, referred to as 'fishers,' were involved in diverse fishing-related activities, including fish catching, post-harvest processing (drying, smoking, marketing), equipment maintenance (boat repairs, net mending), and fish portering.

## Study setting

This study was conducted among 385 artisanal marine fishers in the fishing community of Elmina. Elmina, the administrative capital of KEEA Municipality, is recognized as one of the first European settlements in West Africa. Located 12 km west of Cape Coast, the regional capital of the Central Region, Elmina sits along Ghana's Atlantic coast and covers an area of 452 km² with a population density of 319.8 persons/km² [32]. As Ghana's second-largest fish landing point, Elmina features two landing quays and a bustling fish market. This area was chosen for the study due to its extensive berthing and landing facilities, which attract fishers from across Ghana and other West African countries. It provides an ideal setting for observing settlement patterns, social infrastructure, living conditions, and various aspects of fishing-related activities.

## Data collection

The data collection, which was part of a broader study on fishers in Elmina, Ghana, took place from July to August 2017 [25]. The broader study involved other subjects, including the living conditions of fishermen, their mobility trajectories, living arrangements, knowledge, attitudes, and behaviors regarding HIV, engagement in risky behaviors, and participation in HIV education programs.

**Quantitative data collection.** A semi-structured questionnaire was used to collect data from respondents (S1 Appendix). The semi-structured questionnaire covered various topics, including but not limited to mobility and settlement patterns and HIV risks, HIV and AIDS-related knowledge and attitudes, and risky sexual behaviors. However, the current study concentrated primarily on fisher participation in fish for sex, pregnancy, and birth from FFS relations, as well as associated parenting factors.

**Qualitative data collection.** Interview guides were developed to collect data from key informants and FGD participants (S2 Appendix). The interviews and discussions were conducted in both English and the local languages

of Fante and Twi. For the KIIs conducted in various administrative offices of the KEEA and NGOs, English was the predominant language used. All two FGDs were conducted in Fante, led by the first author, who is fluent in both Fante and Twi. Interactions with key members of fishers' associations were also conducted in Fante and Twi. Each KII lasted between 30–45 minutes, while the FGDs ranged from one hour to one and a half hours. The KIIs and FGDs aimed to gather insights on fishing activities, community challenges, and the living conditions of fishers in and around Elmina. Participants were chosen based on their knowledge and ability to provide essential information pertinent to the research questions.

Participation in the study was voluntary, and all participants provided written informed consent prior to being interviewed (S3 Appendix). The study adhered to all ethical guidelines outlined in the Helsinki Declaration. Moreover, this study was reviewed and approved by the Ethics Committee for the Humanities at the University of Ghana (ECH 118/16–17). All data collected from respondents were stored on a password-protected laptop, which can only be accessible by the first author and authorized persons.

### Data analysis

**Quantitative data analysis.** Descriptive statistics (frequencies and percentages) were used to describe the quantitative data with the aid of the Statistical Package for Social Sciences version 27. We employed descriptive statistics (frequency and percentage) to describe the socio-demographic characteristics of respondents and the frequency of occurrence of FFS related female single parenthood in Elmina. The quantitative data is attached as S4 Appendix.

**Qualitative data analysis.** The qualitative data was analyzed following Braun and Clarke's reflexive thematic analysis (RTA) six distinct phases [33]. The RTA framework allows for an interpretation of the collected data while acknowledging the subjectivity of the researchers' perspectives. We captured, transcribed, and classified our data based on relevant issues identified in the study. To familiarize ourselves with the data, we carefully read the transcripts of respondents. To enhance the study, we produced narrative descriptions that illustrated the connections within the data, quoting statements from focus group discussions (FGDs) and key informant interviews (KIIs) in specific contexts.

The data analysis was conducted collaboratively by both authors, with the first author focusing primarily on the qualitative data and the second author handling the quantitative data. The themes for thematic analysis centered on the frequency of FFS-related female single parenting, the drivers of single parenthood in FFS relationships, and the effects of single female parenting.

Since this study used a convergent parallel mixed-methods design, we merged the quantitative and qualitative findings to compare them. The qualitative data is attached as S5 Appendix.

### Results

#### Socio-demographic characteristics of fishers

Table 2 summarizes the socio-demographic profile of the 385 fishers participating in this study. More than half of fishers (51.4%) were female, while about half were aged 35–64 (49.9%). Most fishers were currently married (55.6%), had formal education (66.5%), and were religiously affiliated (92.5%). Eight out of ten participants (82.1%) identified as Christians, while less than one-tenth of the participants (7.5%) reported having no religious affiliation. Additionally, 38.7% of respondents were fishers engaged in post-harvest activities (processing, marketing, storage, transportation, etc.), with a significant representation of females. One-fourth of the fishers (25.5%) belonged to the Fish Catch Group, which consisted solely of males who venture out to sea to catch fish, as females are restricted from participating in fishing activities in the study area. Furthermore, less than one-fifth of the fishers (14.3%) were part of the Maintenance and Repair Group, responsible for repairing boats, mending gear such as nets, and performing patch repairs on bowls and pans for female fish traders.

**Table 2. Socio-demographic characteristics of fishers.**

| Socio-demographic characteristics | Frequency | Percent |
|---|---|---|
| **Sex** | | |
| Male | 187 | 48.6 |
| Female | 198 | 51.4 |
| **Age** | | |
| <35 | 179 | 46.5 |
| 35–64 | 192 | 49.9 |
| 65+ | 14 | 3.6 |
| **Education** | | |
| No formal education | 129 | 33.5 |
| Had formal education | 256 | 66.5 |
| **Religion** | | |
| Christian | 316 | 82.1 |
| Islam | 21 | 5.5 |
| African Traditionalist | 19 | 4.9 |
| No religion | 29 | 7.5 |
| **Marital Status** | | |
| Never married | 112 | 29.1 |
| Currently married | 214 | 55.6 |
| Formerly married | 59 | 15.3 |
| **Type of fishing occupation** | | |
| Fish Catch Group | 98 | 25.5 |
| Post-harvest Group | 149 | 38.7 |
| Maintenance and Repair Group | 55 | 14.3 |
| Porters and Errand Group | 83 | 21.6 |
| **Total** | **385** | **100.0** |

### Frequency of FFS-related female single parenting

Respondents were asked to determine the frequency of FFS-related single parenting occurrence within the Elmina fishing community, and Table 3 presents the results. Among the 385 fishers, 36.9% indicated that FFS-related single parenthood never occurs. About 19% of fishers reported that FFS-related single parenting rarely happens, while about 3 out of 10 fishers (29.6%) responded that FFS-related single parenting sometimes happens. A little over one-tenth of fishers (11.4%) reported that FFS-related single parenting often happens.

### Drivers of single parenthood in FFS relationships

The study also sought to explore the connection between FFS relationships and single parenting among female FFS participants. Female FFS partners were defined as individuals who had engaged in FFS within the past year. Focus Group Discussions (FGDs) with both male and female participants, along with Key Informant Interviews (KIIs), identified four primary factors contributing to single parenting among women involved in FFS (Fig 1). These drivers included (i) uncertain paternity resulting from multiple sexual partners, (ii) male partners' denial of paternity due to suspicions of promiscuity, (iii) transient fishers who become untraceable sexual partners, and (iv) neglect of children by fathers engaged in multiple partnerships.

**Uncertain paternity resulting from multiple sexual partners.** The FGDs and KIIs revealed that some women who had children with fishers after engaging in FFS become single parents when they were unable to confirm the father of their children due to having multiple FFS partners. Some participants narrated the following to support the theme:

**Table 3. Frequency of occurrence of FFS female single parenthood in Elmina.**

| Response | Frequency | Percent |
|---|---|---|
| Never | 142 | 36.9 |
| Rarely | 72 | 18.7 |
| Sometimes | 114 | 29.6 |
| Often | 44 | 11.4 |
| Always | 13 | 3.4 |
| **Total** | **385** | **100.0** |

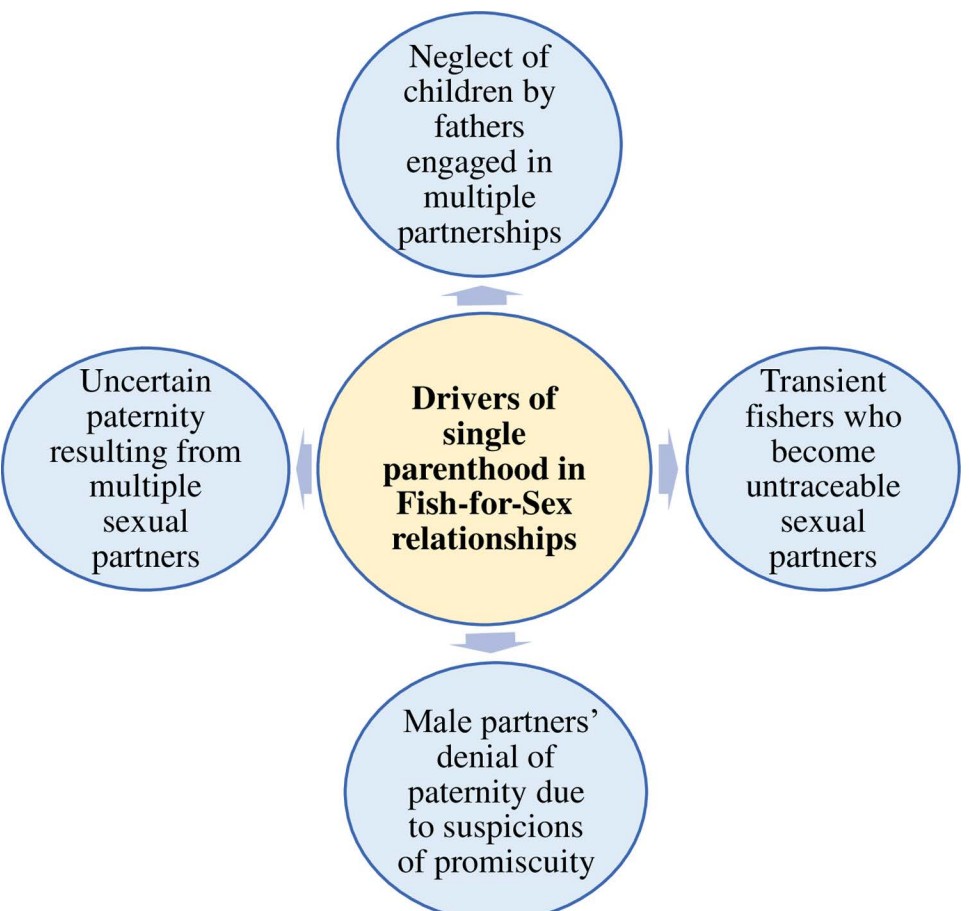

**Fig 1. Drivers of single parenthood in FFS relationships.**

*Some women who may have had multiple sexual partners and are unsure of the father's identity by the time they get pregnant. Consequently, they may choose to assume full responsibility for the child without considering the father's identity.* (KII 1, Municipal Health Directorate)

*I find it difficult to comprehend the girls' actions. How can one woman have three or more boyfriends in this small town? This is why many of them do not know the fathers of their children. All they care about is having sexual relations with fishermen after their expedition, a time when they have a lot of money to spend on women.* (FGD 1, Female fisher)

**Male partners' denial of paternity due to suspicions of promiscuity.** Interviews with key informants and FGD participants revealed that some male FFS partners deny paternity of their children on suspicion of their FFS partners' promiscuity, leading to single female parenting. Some participants shared the following insights:

*Many women in the fishing community are unmarried and have no marital commitments, allowing them to have as many partners as they desire despite societal disapproval of such behavior. Due to this, some male fishers are aware that their partners have other partners and yet engage in sexual relations with them for sexual gratification. However, when unfortunate and unexpected pregnancies occur, some male fishers deny responsibility, accuse the women of being promiscuous, and urge them to blame their other boyfriends for the pregnancy. This often leads to single female parenting as the males refuse to take responsibility for the pregnancy and the child that comes out of it.* (FGD 2, Male fisher)

*My boyfriend accused me of having another man several times, which led to a lack of trust in our relationship. Since then, he has not provided for me and the child. Because he has neglected his responsibilities, I am raising my child alone, even though he knows they look alike and that he is his.* (FGD 3, Female fisher)

*Single parenting is very common in the community. Through radio and community awareness campaigns, we provide advice on protective measures, yet only a small number of individuals heed our guidance. No matter what you tell them, many fishermen refuse to use condoms. Additionally, male fishermen often have multiple female partners, and some fish traders will engage in sexual relations with them if they have access. When these women become pregnant, the men often claim they bear no responsibility.* (KII 2, Community-Based Organisation).

**Transient fishers who become untraceable sexual partners.** The KIIs also found that some FFS men are migrants who have brief sexual encounters, move, and sometimes never return to the community of the sexual encounter, leaving them untraceable if a woman becomes pregnant. Consequently, these women become single parents. A key informant had this to say:

*In this community, most fishermen are migrants from other villages. They merely trade fish and leave the community. After exchanging FFS with the women, they go before the women get pregnant. Occasionally, authorities track down some of the migrant fishers and establish settlements. However, many never return and may not even know they have children in the fishing community they had the sexual encounter.* (KII 2, Department of Fisheries).

**Neglect of children by fathers engaged in multiple partnerships.** In Elmina, some male fishers follow a cultural norm of having multiple wives and additional FFS partners, leading to large families that are difficult to support. The male fishers often struggle to provide for all their children, including those from FFS relationships. Consequently, female FFS partners are left to care for their children independently. A key informant reported the following:

*In some FFS relationships, women transition into the role of a wife without any formal marriage rites. Often, a woman is simply considered a wife by virtue of entering a sexual relationship with a man, which creates an informal partnership. This lack of formal commitment can complicate family dynamics and frequently leads to single parenting among women involved in FFS relationships. Many fishermen in these situations have multiple children across different households, making it difficult for them to provide adequate care for each one.* (KII 2, Community-Based Organisation).

## Effects of single-female parenting

Interviews with participants revealed that FFS has several adverse effects on parenting. These include paternal absence, child developmental challenges, maternal burden, and intergenerational cycle of FFS relationships (Fig 2).

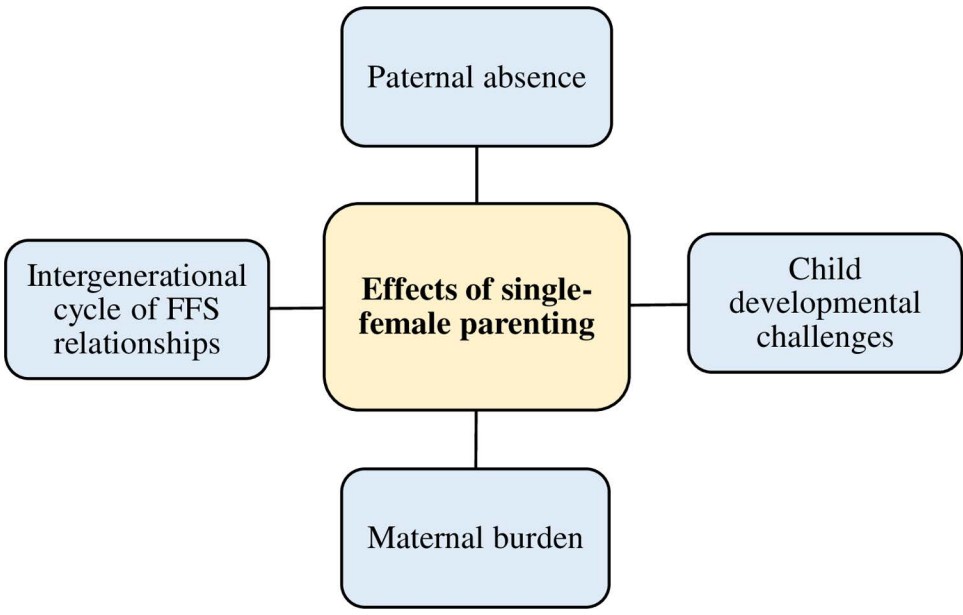

**Fig 2. Effects of single-female parenting.**

**Paternal absence.** Key informants explained that children born out of FFS relationships often grow up without knowing their fathers, leaving them with fragmented family identities and limited guidance, especially in fishing communities. A key informant from the Fisheries Department reported that:

*Many children in Elmina are unaware of their parents' identities. If you go to the town right now and talk to many of them, they will tell you that they do not have fathers. While this may be disheartening, it is a reality within the fishing community.* (KII 2, Department of Fisheries)

*Women who have sex with fishers and cannot determine the paternity of their children or identify their fathers are unable to tell their children who their fathers are, leaving these children to grow up without knowledge of who their fathers are. Many such children are found in fishing communities.* (KII 5, Community Member)

**Maternal burden.** Both the male and female FGD participants highlighted that women who gave birth as a result of FFS relationships face the challenges of pregnancy and childcare alone, often without support, which restricts their ability to provide adequate care for their children and themselves. Participants expressed the following:

*Most women involved in FFS relationships who become pregnant are often unable to locate their fathers or face rejection from them. As a result, they are left to manage their pregnancies and childcare alone, making it extremely challenging to provide adequate care for themselves and their children.* (FGD 6, Female fisher)

*Often, they (women) do not listen when you advise them, and when the men (fishers) refuse their pregnancies, then they rush to Social Welfare to complain. There are instances where we can invite some of the men to make settlements, but in many cases, we are unable to trace the men. As a result, the women struggle to support themselves, with their already impoverished family members unable to assist them.* (FGD 7, Male fisher)

Also, KEEA officials elaborated that children frequently end up with grandparents as mothers struggle to cope, leading to inconsistent care and early self-supporting behaviors. Two officials shared the following:

*With fathers denying paternity and abandoning their children, these children frequently end up living with their grand-parents. Mothers grappling with anger and hardship may struggle to care for them effectively. Consequently, children born into FFS relationships face significant challenges, including inadequate parental care and early sexual relation-ships as a means of self-support.* (KII 10, KEEA official)

*Regrettably, when fathers fail to identify or accept responsibility for their children's upbringing, their mothers also experience emotional disengagement from the children because of the father's actions, leading to inadequate care. Frequently, grandparents take care of the children, but due to their age and insufficient income, they often struggle to provide adequate support.* (KII 9, KEEA official)

**Intergenerational cycle of FFS relationships.** During the FGDs, it emerged that daughters born out of FFS relationships may repeat similar behavior. Such children may also engage in FFS relationships when they grow up, which perpetuates the vicious cycle of FFS relationships and its adverse effects. Some participants explained this situation:

*Some girls involved in FFS relationships within the community are daughters of women who have previously engaged in FFS. Often, these girls do not know their fathers and lack parental guidance. This lack of guidance allows them to act freely, usually associating with male fishers and engaging in similar relationships.* (FGD 8, Female fisher).

*Honestly, parents who had their children through FFS relationships naturally put serious impediments in their lives. Since these births typically stem from a lack of love, the children often feel burdened and learn that their parents do not care about them. In extreme cases, these children may rebel by engaging in risky sexual lifestyles such as FFS and commercial sex work.* (FGD 9, Male fisher).

**Child developmental challenges.** Interviews with key informants noted that the absence of a father figure creates emotional and social obstacles, hindering children's identity formation and sense of belonging as they grow up. This consequently affects their development, especially their social and emotional development. A key informant indicated that:

*Children from FFS relationships who grow up without a paternal figure are often raised by their mothers or grandpar-ents. The absence of a father figure can lead to emotional and social challenges as they navigate their formative years. Without the guidance and support typically provided by a father, these children may struggle to develop a sense of identity and belonging within their communities.* (KII 1, Community-Based Organisation)

## Discussion

The study examined fishers perception of the frequency of FFS-related female single parenthood and the drivers and effects of single parenthood arising from FFS relationships within the Elmina fishing community. Our findings reveal that while FFS-related single parenthood is not universally experienced, 63.1% of fishers acknowledged its occurrence ("Rarely," "Sometimes," "Often," or "Always"). The data highlights the existence of FFS-related single parenthood in Elmina, emphasizing the need for targeted interventions to address its occurrence to enhance the well-being of women and their children.

Prior research indicates that women's economic dependence on fishers in fishing villages frequently results in several concurrent partnerships, subsequently leading to circumstances where women may become single parents due to uncertain paternity [25]. Our findings indicate that sexual relationships with many FFS partners result in women bearing sole

responsibility for their children when paternity is questionable. The situation aligns with research on transactional sexual relationships, wherein economic demands affect relationship decisions, occasionally leading to paternity uncertainty and subsequent single parenting [34]. Evidence suggests these partnerships frequently lack openness and enduring commitment, complicating the establishment of paternal responsibility [35].

The results show that suspicion of promiscuity within FFS relationships also contributes significantly to single parenting. The FGDs highlight that men who are aware of their partners' sexual involvement with multiple fishers may distrust the fidelity of their relationships, leading them to reject paternity claims [36]. In such cases, women face economic and social hardships, raising children independently in the absence of financial or emotional support from their partners. Studies of transactional relationships in economically marginalized settings have documented similar patterns where men deny paternity to avoid financial and social responsibilities, often placing the full burden of parenting on women [37]. This lack of support contributes to the feminization of poverty, exacerbating gender disparities in these communities [38,39].

Furthermore, the results show that when women in FFS relationships maintain relationships with multiple fishers, they often face challenges identifying the biological father of their children, leading to single parenting. This trend reveals the complexities of transactional relationships; women who are dependent on multiple partners for economic support may prioritize financial gain over establishing stable relationships [40–42]. The drivers of FFS are multifaceted, encompassing environmental and socio-economic factors that create an environment where women feel compelled to engage in such relationships [20,24–26]. Moreover, to place this study within a wider discourse on FFS, it is essential to recognize that similar patterns of economic necessity driving women into transactional relationships have been documented in various contexts beyond Ghana, particularly in Malawi, Kenya, Tanzania, Nigeria, and Zambia, where socio-economic instability and gender inequalities persist [18,19,24,43]. In communities that morally judge promiscuity and infidelity, uncertain paternity reinforces social stigma and prompts some women to care for their children alone to avoid confrontation and further social scrutiny. This moral judgment can exacerbate feelings of isolation and helplessness among single mothers, making it difficult for them to seek help or build supportive networks. Consequently, these dynamics not only affect individual families but also contribute to broader societal issues, such as poverty and reduced access to resources for children.

The migration patterns of fishers, who often move from village to village, exacerbate the issue of uncertain paternity. Migrant fishers typically engage in short-term sexual relationships with women in each community [10] with little intention of maintaining a lasting presence or providing ongoing support if pregnancies occur. The transient nature of FFS relationships, as revealed by the results, presents a significant challenge for women, often leaving them to raise children alone when fishers fail to return. Other contexts, where migratory labor contributes to family fragmentation and leaves women as single parents due to the absence of stable male partners [40], witness similar patterns. These dynamics strain the emotional and financial resources of these women and affect the well-being of the children, who may grow up without both parents' guidance and support. As a result, the cycle of instability can perpetuate, affecting future generations and raising concerns about the long-term social implications of such transient relationships.

Cultural norms permitting polygamous and casual partnerships among male fishers further complicate family structures in fishing communities. In these relationships, some women informally transition into quasi-marital roles without formal commitment, leaving them vulnerable when men, already financially stretched by other partnerships, are unable or unwilling to provide for additional children. In patriarchal societies, where men often distribute limited resources across multiple households, this form of relationship dynamic results in women bearing the primary responsibility for childcare [44]. The economic implications for women are dire, as they struggle to meet their children's needs without adequate support, contributing to a cycle of poverty and marginalization.

Children born from FFS relationships who single mothers or grandparents raise face significant developmental (social and emotional) challenges that can impact their well-being and future choices. Without the presence of a father figure, these children often lack the stability, support, and structure that can provide essential guidance during their formative years. Research suggests that the father's absence is linked to a range of socio-emotional and developmental

issues, including difficulties in forming a stable identity and a heightened risk of behavioral problems, especially as children transition into adolescence [45,46]. The absence of paternal guidance often leaves these children feeling isolated and uncertain of their place in their communities, impacting their ability to build secure relationships and a sense of belonging.

For mothers involved in FFS relationships, raising children alone without the support of the fathers often leads to a multitude of challenges. Emotionally and financially strained, these women struggle to provide a nurturing environment, which can negatively affect their children's development. Studies have shown that children of single mothers in economically unstable conditions often experience poorer health outcomes, reduced educational attainment, and difficulties in psychosocial development [47,48]. The stress of unsupported motherhood in these communities may prevent these mothers from meeting their children's needs fully, contributing to cycles of poverty and limited opportunities.

In cases where children are cared for by grandparents, often due to paternal denial of responsibility or the mother's difficulties in managing single parenting, the children may experience inconsistent caregiving. This finding supports previous studies that found that grandparents provide care to their grandchildren when there is parental abandonment [49,50]. Children being cared for by their grandparents may perceive this arrangement as a form of abandonment, which may lead to behavioral and emotional challenges. This dynamic is further complicated when children witness or experience neglect, resulting in an increased likelihood of early relationships and behaviors that mirror their unstable environment. Such patterns echo findings that children from low-support environments are more vulnerable to engaging in early sexual relationships and other high-risk behaviors as a means of seeking security and self-worth [51].

A striking aspect of the FFS context is the intergenerational nature of the practice, wherein daughters of previous FFS participants are often drawn into similar arrangements. Lacking knowledge of their fathers and traditional family boundaries, these girls may engage in FFS relationships with fishers, perpetuating a cycle that intertwines economic need with limited emotional support. This cyclical vulnerability reflects the constrained choices available to these young women and underscores the impact of an absent paternal figure on the structure and function of the family unit [40]. The absence of consistent parental figures reinforces a socio-economic environment that leaves these children with few alternatives for building stable, self-sufficient lives.

Ultimately, children from FFS relationships often navigate an unstable socio-economic and emotional landscape that can affect their mental health, academic potential, and future relationships. The generational cycle of FFS reflects broader societal and community challenges, as economic vulnerability and gendered power imbalances continue to limit the choices available to women and children in these communities. Addressing the drivers of FFS, including political, environmental, and socio-economic factors, is crucial for developing effective interventions. Efforts to break this cycle require a focus on holistic support systems, from community education and economic initiatives to mental health resources, aimed at providing children with the foundational support they need for healthier, more secure futures.

## Limitations of the study

The study has some limitations worth noting. First, as a cross-sectional study, it cannot establish causality. Also, the findings rely on reports from community members rather than from women who are single parents because of FFS relationships, which may limit the study's ability to capture the complete reality of the issue. Future research should interview women who have children from FFS relationships so their perspectives can be captured to help in understanding the nuances of FFS relationships. Nonetheless, these limitations do not detract from the value of the current study, which makes a significant contribution to the limited knowledge of the drivers and effects of FFS-related single parenthood.

## Implications for policy and practice

This study's findings have major policy and practice implications, particularly in addressing the vulnerabilities of women and children in the Elmina fishing community. Given that FFS-related single parenting is a recognized phenomenon,

interventions should aim on lowering economic dependency and enhancing social protections for affected women and children.

Policy solutions should prioritize economic empowerment programs for women who are in FFS relationships. Financial independence programs, such as vocational training, microfinance possibilities, and cooperatives for female vendors, can help them rely less on transactional sexual relationships for economic survival.

Strengthening legal frameworks around paternity recognition and child support is crucial. Policymakers must advocate to enforce child support laws that hold fathers accountable, ensuring that children born into FFS partnerships receive enough financial and emotional assistance. Community-based advocacy and legal literacy initiatives can help women seek justice in cases of paternal neglect. Social and behavioral therapies are equally important. Community education programs should challenge cultural practices that stigmatize single mothers while also encouraging responsible fatherhood. Engaging fishermen, community leaders, and traditional authorities in discussions about ethical fishing and responsible relationship behaviors could help to create more stable family systems. Furthermore, psychological and social support services should be enhanced to address the emotional and mental health needs of single mothers and their children.

Finally, specific treatments for children in FFS partnerships should include educational support, mentorship programs, and access to social welfare services. Addressing intergenerational vulnerabilities will be critical in breaking the cycle of economic and social deprivation caused by FFS connections in the Elmina neighborhood.

## Conclusions

Overall, FFS relationships contribute to single parenting, uncertain paternity, and inadequate child support in fishing communities. These findings underscore the need for targeted interventions that promote awareness, encourage protective measures, and provide economic support for single mothers. Children born from FFS relationships often encounter significant emotional and social difficulties due to absent or unacknowledged fathers and limited parental guidance. Children born from FFS relationships raised by single mothers or grandparents may face identity and belonging challenges, lacking the stability and support essential for healthy child development.

The burden on single mothers, who must manage caregiving alone, further reduces the structure and support these children receive, potentially driving some toward early relationships or self-sustaining behaviors. This pattern, often perpetuated across generations, illustrates how daughters of FFS participants may follow similar paths, highlighting the intergenerational impact on community stability and child well-being.

These findings reveal a complex interplay of economic pressures, cultural norms, and gendered power imbalances that influence family structures and caregiving roles within these fishing communities. Addressing these challenges will require both community-centered and policy-driven initiatives that support single mothers and ensure children have consistent access to essential resources and support systems.

## Supporting information

**S1 Appendix. Questionnaire for fishers.**
(DOCX)

**S2 Appendix. Interview guides for participants.**
(DOCX)

**S3 Appendix. Protocol consent form.**
(PDF)

**S4 Appendix. Quantitative data.**
(XLSX)

**S5 Appendix. Qualitative data.**
(DOCX)

## Acknowledgments

We are grateful to all the participants in this study.

## Author contributions

**Conceptualization:** Sylvester Kyei-Gyamfi, Frank Kyei-Arthur.

**Formal analysis:** Sylvester Kyei-Gyamfi, Frank Kyei-Arthur.

**Investigation:** Sylvester Kyei-Gyamfi.

**Methodology:** Sylvester Kyei-Gyamfi.

**Supervision:** Sylvester Kyei-Gyamfi.

**Validation:** Sylvester Kyei-Gyamfi, Frank Kyei-Arthur.

**Visualization:** Frank Kyei-Arthur.

**Writing – original draft:** Sylvester Kyei-Gyamfi, Frank Kyei-Arthur.

**Writing – review & editing:** Sylvester Kyei-Gyamfi, Frank Kyei-Arthur.

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
