## [Decision Letter · Decision Letter 0]

PONE-D-24-50030Drivers and effects of fish-for-sex related single parenthood in a fishing coastal community in GhanaPLOS ONE

Dear Dr. Kyei-Arthur,

Thank you for submitting your manuscript to PLOS ONE. After careful consideration, we feel that it has merit but does not fully meet PLOS ONE’s publication criteria as it currently stands. Therefore, we invite you to submit a revised version of the manuscript that addresses the points raised during the review process.

We look forward to receiving your revised manuscript.

Kind regards,

Juan Carlos Rocha Gordo

Academic Editor

PLOS ONE

Additional Editor Comments:

Dear authors,

It took over 30 invitation to reviewers to secure the 2 reviews attached. Please take their feedback seriously. Your study is currently lacking a number of features to make it reproducible, including ethics approval, consent forms, interviews and questionnaire protocols, to name a few. The reviewers have pointed out sections of the paper that lack clarity, for example where it is unclear if your results apply to men, women or both. Given the sensitivity of the topic you're dealing with, an outmost attention to the detail is required, both in language and transparency with the methods and procedures performed on the field. Be clear when a result has internal or external validity(if it applies to a town, a region, or is generalizable to other social contexts e.g. other countries). I hope you find the reviewers comments constructive and help you improve the quality of your work.

Best,

Juan Rocha

Reviewers' comments:

Reviewer's Responses to Questions

**Comments to the Author**

1. Is the manuscript technically sound, and do the data support the conclusions?

Reviewer #1: Yes

Reviewer #2: Partly

2. Has the statistical analysis been performed appropriately and rigorously? 

Reviewer #1: Yes

Reviewer #2: I Don't Know

3. Have the authors made all data underlying the findings in their manuscript fully available?

Reviewer #1: Yes

Reviewer #2: No

4. Is the manuscript presented in an intelligible fashion and written in standard English?

Reviewer #1: Yes

Reviewer #2: Yes

5. Review Comments to the Author

Reviewer #1: Thank you for carrying out such an interesting study. However, I have some concerns.

1. The author(s) should kindly number the equation presented in their study.

2. The initial section of the study should fully present the abbreviation KEEA before using it. (see Line 76).

3. The author(s) indicate that a convergent parallel mixed-method approach was used in their study, but this is

not really seen in their work. It will be important for the author(s) in the material and methods section to

clearly show how the following were carried out under separate sections:

a. Quantitative design and methods—study design, population/sample was carried out.

b. Qualitative design and methods—a detailed write-up on the participants, the interview process, focus

group discussion, and how the results were analyzed.

c. The same approach in point b should be followed in the results section. The authors should clearly

present the findings from the quantitative approach as well as those from the qualitative approach.

4. In lines 97-99, the sentence is not clear. What ten associations are the author(s) referring to?

5. What is the composition of the focus group discussions?

6. The author(s) should clearly present the theoretical and practical implications of their work as well as future

directions.

Reviewer #2: The migration of fishers from one community to another is often associated with fish-for-sex (FFS) exchanges. = really is it? like all over the world or just in Ghana or where? I don't know if that stands true in like Ireland or Sweden for example so maybe you could nuance that first sentence

convergent parallel mixed-method approach= what is this? can you use accessible language so we can understand what you mean or else explain this in brackets so you don't have the audience confused

while the transcript of participants was analyzed thematically. = surely it was many transcripts?

that most fishers (63.1%) indicated the occurrence of FFS female single parenthood in Elmina. = I dont get it, you asked fishers (men?) if there was FFS in Elmina and 63.1% said yes? But they aren't single females with kids right?

Most fishers do not limit their work to a single fishing community. = on the world? which fishers? You mean to say that fishing and trade is transboundary? Or to hint to that?

To catch, trade, or engage in any fishing-related activity, it is essential to be located within or near a fishing area= this is a bit of a weird sentence? You could trade dry fish but not be near a "fishing area" at all...

Men typically move between fishing communities to perform off-shore fishing activities (such as catching fish at sea or working on boats and other equipment). In contrast, women often perform near-shore fishing activities (such as accompanying them to buy or sell fish across different fishing locations) [1-4]. = you need to give us the geographies where this holds true? are you talking about small-scale fisheries in Ghana or commercial fleets in russia or what? Women don't just "accompany" men to buy or sell fish.... Women in the industrial Ghanaian Tuna value chain finance large commercial vessels and are owners of vast trade businesses. I would nuance your sentence, the next sentence is great, and I would draw on more of the gender in fisheries work that for example GAF (gender in fisheries and aquaculture) members have done, you could reference more of them especially in the following sentence about interdependencies.

Risky sexual behaviour among fishers, such as multiple sexual partners and non-condom use [11-13], can lead to pregnancy and increase the risk of sexually transmitted infections for both fishers and their partners. = so the women partners are not fishers? I think you could clarify earlier who the "women" are as the men are given an identity as "fishers". Are they traders for example?

Also, women who engage in sexual intercourse with multiple partners may be uncertain about the paternity of their children [14]. = but also men with multiple partners wouldn't know if they have children and where, so its more balanced sentence.

Despite its far-reaching impact on women, this issue has received limited attention in fisheries literature, particularly in Ghana. = on women but also the children and it does not have an impact on men then?

KEEA - is this an acronym?

Qualitative methods allowed for an in-depth examination of fishers' mobility, safety concerns, and experiences of sexual harassment, = im still confused as to who the fishers are, are they the men only? so the lads were getting harassed by the women for sex or?

The number of fishers in Elmina is unknown = I would say why e.g. due to lack of government interest in small-scale artisanal fishers or lack of national statistics etc. Its not unknown for no reason, its political

I would contextualize this study earlier by describing the fisheries and the communities in one or two sentances in the intro- like are we talking about small-scale fisheries? or big industrial bait boats or? The reader could then much easier picture the context. And if you are talking about SSF then you can draw on all the references for that work in Ghana and west Africa and globally that highlights their importance but complete lack of political support- hence why we end up with women having to do what they do to survive. Right now the paper up till now is a bit "dry", we dont know the who and you don't situate the paper very well yet in the wealth of literature on SSF or gender in fisheries.

Table 1: Participants of the Key Informant Interviews = We need to know the genders so as readers we can be sure you didn't talk to a bunch of men about women's issues.

The same goes for the quantitative data who were the respondents? We need an overview of their profiles, ages and genders. It seems you only interviewed "fishers" ? so only men?? what about the female fish traders or the women who are getting the FFS??

I need more details on the ethics- this are very sensitive topics and information- we need more information on how your participants gave consent and a copy of the consent form in appendix and the plain language statement that informed them of the project and all the details i.e. what you would do with their data, data storage. We also need a lot more details on participant recruitment, for all sectors not just fishers.

Data analysis- is one paragraph? we dont even know the language the interviews and discussions were done in? who did the analysis? we need a lot more details on this step... what were the themes for analysing thematically? how was this done? where? What are narrative descriptions? how were they chosen?

Results= ok now we are getting the genders and the who of the people- I would put this way before as the reader goes through the whole first part not know "who"

Table 2= needs more explanation- what is a fish catch group, what gears are used by what genders, what are the main species? what religions? what does the post-harvest group do?

table 3= so it wasnt ffs female headed households you were talking to? it was just fishers estimating if they know any single women with kids that come from FFS? i don't fully get it... what about the single women with kids? were they participants and if not why?

Focus Group Discussions (FGDs) with both male and female participants = at the same time???

I find it difficult to comprehend the girls' actions. How can one woman have three or more boyfriends in this small town? This is why many of them don't know the fathers of their children. All they care about is having sexual relations with fishermen after their expedition, a time when they have a lot of money to spend on women. (FGD 1, Female fisher) = but i thought they were having sex to get the fish to trade or for home consumption? How is that not understood? In fact I think FFS needs more explanation than one sentence in the intro- what is the fisher used for where does it go and what species fresh or then dried here in elmina? and why is it done? why do women have sex for fish- like this needs to be properly presented so it doesnt look like they are just out having multiple partners for fun.

Many women in the fishing community are unmarried and have no marital commitments, allowing them to have as many partners as they desire despite societal disapproval of such behaviour. = but they are trying to get fish to sell or feed their kids right?? These quotes are putting a lot of blame on women without a proper explanation of why FFS

The discussion could do with citing of literature beyond Ghana and placing the study in the global discourse on FFS or even branching out to other products as there is a lot out there, but its very interesting to take the impacts on the kids into account, thats a power of this study. And those findings are discussed at a beyond ghana level so thats super interesting to read, about the generational issues of FFS. But i think the discussion needs to widen and also make some higher level links to the drivers of FFS e.g. politics, environmental change, lack of socio-economic supports in SSF etc.

6. PLOS authors have the option to publish the peer review history of their article (what does this mean? ). If published, this will include your full peer review and any attached files.

**Do you want your identity to be public for this peer review?** For information about this choice, including consent withdrawal, please see our Privacy Policy .

Reviewer #1: No

Reviewer #2: No

---

## [Author Response · Author response to Decision Letter 1]

5 Mar 2025

Responses to reviewers’ comments

Journal requirements

Response: The manuscript meets PLOS ONE's style requirements.

Response: We will change the data availability statement to “There are restrictions on sharing of these data since they contain sensitive data. Interested persons can obtain the data from the corresponding author upon reasonable. Dataset requests may be sent to Frank Kyei-Arthur (fkyei-arthur@uesd.edu.gh).”

Response: We have included a separate caption for each figure in the revised manuscript. Please refer to lines 255 and 320.

Reviewer #1

1. Thank you for carrying out such an interesting study. However, I have some concerns.

Response: Thank you for the commendation. We appreciate it.

2. The author(s) should kindly number the equation presented in their study.

Response: We have numbered the equation in the revised manuscript. Please refer to lines 116-119.

3. The initial section of the study should fully present the abbreviation KEEA before using it. (see Line 76).

Response: KEEA means Komenda-Edina-Eguafo Abrem. It has been inserted in the revised manuscript. Please refer to lines 96-97.

4. The author(s) indicate that a convergent parallel mixed-method approach was used in their study, but this is not really seen in their work. It will be important for the author(s) in the material and methods section to clearly show how the following were carried out under separate sections:

a. Quantitative design and methods—study design, population/sample was carried out.

b. Qualitative design and methods—a detailed write-up on the participants, the interview process, focus group discussion, and how the results were analysed.

c. The same approach in point b should be followed in the results section. The authors should clearly present the findings from the quantitative approach as well as those from the qualitative approach.

Response: We thank the reviewer for the suggestion. We have created sub-sections for quantitative sampling procedure and qualitative sampling procdure to make it easier for readers to sampling procedure for each. Please refer to lines 112-141.

Also, we have created sub-sections for quantitative data collection and qualitative data colection. Please refer to lines 169-191.

Furthermore, we have created sub-sections for quantitative data analysis and qulaitative data analysis. Please refer to lines 193-216.

However, for the results section, we have decided to maintain it as it is since our study used a convergent parallel mixed-method approach and we have merged the quantitative and qualitative findings to compare them. This makes it easier for readers to see the triangulation of the quantitative and qualitative findings. We have stated that in the revised manuscript. Please refer to lines 215-216.

5. In lines 97-99, the sentence is not clear. What ten associations are the author(s) referring to?

Response: An additional sentence has been added to make the sentence clearer than it was in the previous manuscript. Please refer to lines 122-129.

6. What is the composition of the focus group discussions?

Response: The composition has been inserted to make the sentence clearer. Please refer to lines 133-141.

7. The author(s) should clearly present the theoretical and practical implications of their work as well as future directions.

Reponse: Thank you for the suggestion. We have added a new section on implications for Policy and practice, which may serve as a policy guidance for the future. Please refer to lines 508-531.

Reviewer #2

1. The migration of fishers from one community to another is often associated with fish-for-sex (FFS) exchanges. = really, is it? like all over the world or just in Ghana or where? I don't know if that stands true in like Ireland or Sweden for example so maybe you could nuance that first sentence

Response: Suggestion to nuance the statement is in the right direction. It has been done. Please refer to lines 69-71.

2. Convergent parallel mixed-method approach= what is this? can you use accessible language so we can understand what you mean or else explain this in brackets, so you don't have the audience confused

Response: Additional narrative has been added to explain the use of convergent parallel mixed method approach. Please refer to lines 103-106.

3. While the transcript of participants was analysed thematically. = surely it was many transcripts?

Response: Yes, we analyzed all transcript thematically, consisting two FGD transcripts and 30 KII transcripts.

4. That most fishers (63.1%) indicated the occurrence of FFS female single parenthood in Elmina. = I don’t get it, you asked fishers (men?) if there was FFS in Elmina and 63.1% said yes? But they aren't single females with kids right?

Reponse: The occurrence of FFS female single parenthood was asked all fishers to ascertain the frequency of occurrence of FFS single parenthood in Elmina. FFS female single parenthood talks about single females who had children/kids due to FFS relationships.

5. Most fishers do not limit their work to a single fishing community. = on the world? which fishers? You mean to say that fishing and trade is transboundary? Or to hint to that?

Response: Yes, the statement is general and factual and relates fishing in marine bodies. We have added references to support the claim in the sentence in lines 58-68.

6. To catch, trade, or engage in any fishing-related activity, it is essential to be located within or near a fishing area= this is a bit of a weird sentence? You could trade dry fish but not be near a "fishing area" at all...

Response: Yes, the sentence was problematic and has been deleted.

7. Men typically move between fishing communities to perform offshore fishing activities (such as catching fish at sea or working on boats and other equipment). In contrast, women often perform near-shore fishing activities (such as accompanying them to buy or sell fish across different fishing locations) [1-4]. = you need to give us the geographies where this holds true? are you talking about small-scale fisheries in Ghana or commercial fleets in Russia or what? Women don't just "accompany" men to buy or sell fish.... Women in the industrial Ghanaian Tuna value chain finance large commercial vessels and are owners of vast trade businesses. I would nuance your sentence, the next sentence is great, and I would draw on more of the gender in fisheries work that for example GAF (gender in fisheries and aquaculture) members have done, you could reference more of them especially in the following sentence about interdependencies.

Response: Thank you for the critique. We have taken the issues raised and factored it in the new narrative in sentence in lines 58-68 and supported with relevant references including the GAF as suggested.

8. Risky sexual behaviour among fishers, such as multiple sexual partners and non-condom use [11-13], can lead to pregnancy and increase the risk of sexually transmitted infections for both fishers and their partners. = so the women partners are not fishers? I think you could clarify earlier who the "women" are as the men are given an identity as "fishers". Are they traders for example?

Response: We have made the sentence clearer clarifying who the actors are, either male or female fishers in the new manuscript. Please refer to lines 79-86.

9. Also, women who engage in sexual intercourse with multiple partners may be uncertain about the paternity of their children [14]. = but also men with multiple partners wouldn't know if they have children and where, so its more balanced sentence.

Response: We have replaced the statement with a balanced one which reads ‘Also, both female and male fishers who engage in sexual intercourse with multiple partners may be uncertain about the paternity of their children’ in lines 85-86.

10. Despite its far-reaching impact on women, this issue has received limited attention in fisheries literature, particularly in Ghana. = on women but also the children and it does not have an impact on men then?

Response: The effects its not only on women but parenting in general. The new statement focuses on ‘adverse impact on parenting’ and can be found in line 90.

11. KEEA - is this an acronym?

Response: KEEA means Komenda-Edina-Eguafo Abrem and has been inserted in the revised manuscript in lines 96-97.

12. Qualitative methods allowed for an in-depth examination of fishers' mobility, safety concerns, and experiences of sexual harassment, = I’m still confused as to who the fishers are, are they the men only? so the lads were getting harassed by the women for sex or?

Response: We have revsied the sattement to read ‘Qualitative methods allowed for an in-depth examination of both male and female fishers' mobility, safety concerns, and experiences of sexual harassment, while quantitative methods facilitated the analysis of relationships between the study's key variables’. Please refer to lines 108-111.

13. The number of fishers in Elmina is unknown = I would say why e.g. due to lack of government interest in small-scale artisanal fishers or lack of national statistics etc. Its not unknown for no reason, its political

Response: It is unknown due to the lack of national statistics on small-scale artisanal fishers. We have indicated it in the revised manuscript. Please refer to lines 113-114.

14. I would contextualize this study earlier by describing the fisheries and the communities in one or two sentences in the intro- like are we talking about small-scale fisheries? or big industrial bait boats or? The reader could then much easier picture the context. And if you are talking about SSF then you can draw on all the references for that work in Ghana and west Africa and globally that highlights their importance but complete lack of political support- hence why we end up with women having to do what they do to survive. Right now the paper up till now is a bit "dry", we don’t know the who and you don't situate the paper very well yet in the wealth of literature on SSF or gender in fisheries.

Response: A suitable context has been provided as recommended, and we hope it meets the reviewer's expectations while also enhancing the initial introduction of the paper. Please refer to lines 48-57.

15. Table 1: Participants of the Key Informant Interviews = We need to know the genders so as readers we can be sure you didn't talk to a bunch of men about women's issues.

Response: The sex of the KII participants has been given in the new Table 1 on page 7.

16. The same goes for the quantitative data who were the respondents? We need an overview of their profiles, ages and genders. It seems you only interviewed "fishers" ? so only men?? what about the female fish traders or the women who are getting the FFS??

Response: We have kindly provided a write-up of the socio-demographic characteristics of fishers under the socio-demographic characteristics of fishers sub-section on lines 220-231 and in Table 2 on pages 11-12.

17. I need more details on the ethics- this are very sensitive topics and information- we need more information on how your participants gave consent and a copy of the consent form in appendix and the plain language statement that informed them of the project and all the details i.e. what you would do with their data, data storage. We also need a lot more details on participant recruitment, for all sectors not just fishers.

Response: We have adequately provided information on how respondents were sampled and their data collected. Please refer to lines 112-141.

Also, we have provided additional information on how their data was stored and who had access to them. Please refer to lines 189-191.

Further, we have provided the a copy of the questionnaire, interview guides and consent form as a supplementary materials (S1, S2 and S3 Appendices).

18. Data analysis- is one paragraph? we don’t even know the language the interviews and discussions were done in? who did the analysis? we need a lot more details on this step... what were the themes for analysing thematically? how was this done? where? What are narrative descriptions? how were they chosen?

Response: These concerns relate to data analysis and collection, so we have addressed them suitably under their respective sections. For data collection, please refer to lines 169-191.

For the data analysis section, please refer to lines 193-216.

19. Results= ok now we are getting the genders and the who of the people- I would put this way before as the reader goes through the whole first part not know "who"

Response: Thanks for the commendation.

20. Table 2= needs more explanation- what is a fish catch group, what gears are used by what genders, what are the main species? what religions? what does the post-harvest group do?

Response: The explanation and additional information have been provided to answer the issues raised in lines 223-231.

Additional and more detailed statistics on religion has been added in Table 2 on pages 11-12.

Unfortunately, the study did not inquire into species and so we cannot have that information.

The details on the composition of the fish catch group and the post-harvest group are all detailed out in the narratives in lines 223-231.

21. Table 3= so it wasn’t ffs female headed households you were talking to? it was just fishers estimating if they know any single women with kids that come from FFS? I don't fully get it... what about the single women with kids? were they participants and if not why?

Response: The question was a follow-up inquiry asking all participants about their perception of the frequency of single parenting arising from FFS relations. Consequently, both male and female fishers, as well as those who were married or unmarried but had knowledge of the phenomenon, were required to answer the question.

22. Focus Group Discussions (FGDs) with both male and female participants = at the same time???

Response: No please, two FGDs were held; one for males and the other for females. This is explained in the Methods Section in lines 133-141.

23. I find it difficult to comprehend the girls' actions. How can one woman have three or more boyfriends in this small town? This is why many of them don't know the fathers of their children. All they care about is having sexual relations with fishermen after their expedition, a time when they have a lot of money to spend on women. (FGD 1, Female fisher) = but i thought they were having sex to get the fish to trade or for home consumption? How is that not understood? In fact I think FFS needs more explanation than one sentence in the intro- what is the fisher used for where does it go and what species fresh or then dried here in Elmina? and why is it done? why do women have sex for fish- like this needs to be properly presented so it doesn’t look like they are just out having multiple partners for fun.

Response: The statement came from a female fisher who is clearly opposed to the phenomenon, reflecting a sentiment shared by many women who disapprove of it. The authors of the current paper previously published work on FFS, addressing some related issues; therefore, this study specifically focuses on single parenthood arising from FFS. However, in response to the reviewer comments, we have added more details about fish-for-sex in the introduction section in line 74-78.

24. Many women in the fishing community are unmarried and have no marital commitments, allowing them to have as many partners as they desire despite societal disapproval of such behaviour. = but they are trying to get fish to sell or feed their kids, right?? These quotes are putting a lot of blame on women without a proper explanation of why FFS

Response: Thank you for your opinion, but the authors did not intend to place blame. The quote originated from a community member whose credi

---

## [Decision Letter · Decision Letter 1]

Drivers and effects of fish-for-sex related single parenthood in a fishing coastal community in Ghana

PONE-D-24-50030R1

Dear Dr. Kyei-Arthur,

We’re pleased to inform you that your manuscript has been judged scientifically suitable for publication and will be formally accepted for publication once it meets all outstanding technical requirements.

Kind regards,

Juan Carlos Rocha Gordo

Academic Editor

PLOS ONE

Additional Editor Comments (optional):

Dear Authors,

Both reviewers exprese gratitude for your due diligence in addressing their comments, and both of them recommend acceptance for publication. Congratulations on your work!

Best regards,

Juan

Reviewers' comments:

Reviewer's Responses to Questions

**Comments to the Author**

1. If the authors have adequately addressed your comments raised in a previous round of review and you feel that this manuscript is now acceptable for publication, you may indicate that here to bypass the “Comments to the Author” section, enter your conflict of interest statement in the “Confidential to Editor” section, and submit your "Accept" recommendation.

Reviewer #1: All comments have been addressed

Reviewer #2: All comments have been addressed

2. Is the manuscript technically sound, and do the data support the conclusions?

Reviewer #1: Yes

Reviewer #2: Yes

3. Has the statistical analysis been performed appropriately and rigorously? 

Reviewer #1: Yes

Reviewer #2: I Don't Know

4. Have the authors made all data underlying the findings in their manuscript fully available?

Reviewer #1: Yes

Reviewer #2: Yes

5. Is the manuscript presented in an intelligible fashion and written in standard English?

Reviewer #1: Yes

Reviewer #2: Yes

6. Review Comments to the Author

Reviewer #1: The author(s) have conducted an interesting study, and I would like to thank them for taking the time to address all the concerns raised about their paper. I accept the response they have provided in relation to the concerns I raised.

Reviewer #2: The paper has significantly improved, it's so good now! Reads so well and has all the details I wanted as a reader to understand the study plus the nuances of the gender dimensions and presentation of results. Thanks for addressing everything.

7. PLOS authors have the option to publish the peer review history of their article (what does this mean? ). If published, this will include your full peer review and any attached files.

**Do you want your identity to be public for this peer review?** For information about this choice, including consent withdrawal, please see our Privacy Policy .

Reviewer #1: No

Reviewer #2: No

---

## [Editor Report · Acceptance letter]

PONE-D-24-50030R1

PLOS ONE

Dear Dr. Kyei-Arthur,

I'm pleased to inform you that your manuscript has been deemed suitable for publication in PLOS ONE. Congratulations! Your manuscript is now being handed over to our production team.

Kind regards,

on behalf of

Dr. Juan Carlos Rocha Gordo

Academic Editor

PLOS ONE